

# Use of column experiments to investigate the fate of organic micropollutants – a review

Stefan Banzhaf[1], Klaus H. Hebig[2]

[1]Department of Earth Sciences, University of Gothenburg, Box 460, 405 30 Göteborg, Sweden
[2] Department of Applied Geosciences, Technische Universität Berlin, BH 3-2, Ernst-Reuter-Platz 1, 10587 Berlin, Germany

*Correspondence to*: Stefan Banzhaf (stefan.banzhaf@gu.se)

**Abstract.** Although column experiments are frequently used to investigate the transport of organic micropollutants, little guidance is available on what they can be used for, how they should be set up, and how the experiments should be carried out. This review covers the use of column experiments to investigate the fate of organic micropollutants. Alternative setups are

discussed together with their respective advantages and limitations. An overview is presented of published column experiments investigating the transport of organic micropollutants, and suggestions are offered on how to improve the comparability of future results from different experiments. The main purpose of column experiments is to investigate the transport and attenuation of a specific compound within a specific sediment or substrate. The transport of (organic) solutes in groundwater is influenced by the chemical and physical properties of the compounds, the solvent (i.e. the groundwater, including all solutes),

and the substrate (the aquifer material). By adjusting these boundary conditions a multitude of different processes and related research questions can be investigated using a variety of experimental setups. Apart from the ability to effectively control the individual boundary conditions, the main advantage of column experiments compared to other experimental setups (such as those used in field experiments, or in batch microcosm experiments) is that conservative and reactive solute breakthrough curves can be derived, which represent the sum of the transport processes. There are well-established methods for analyzing

these curves. The effects observed in column studies are often a result of dynamic, non-equilibrium processes. Time (or flow velocity) is an important factor, in contrast to batch experiments where all processes are observed until equilibrium is reached in the substrate-solution system. Slight variations in the boundary conditions of different experiments can have a marked influence on the transport and degradation of organic micropollutants. This is of critical importance when comparing general results from different column experiments investigating the transport behavior of a specific organic compound. Such variations

unfortunately mean that the results from most column experiments are not transferable to other hydrogeochemical environments but are only valid for the specific experimental setup used

Column experiments are fast, flexible, and easy to manage; their boundary conditions can be controlled and they are cheap compared to extensive field experiments. They can provide good estimates of all relevant transport parameters. However, the obtained results will almost always be limited to the scale of the experiment and not directly transferrable to field scales as too

many parameters are exclusive to the column setup. The challenge for the future is to develop standardized column experiments on organic micropollutants in order to overcome these issues.



**Keywords**

Pharmaceuticals, pesticides, column study, groundwater, redox conditions, pH, sorption, xenobiotics, OMPs

## 1 Why we need to investigate the fate of organic micropollutants in groundwater

The presence of organic micropollutants in aquatic environments has been of great concern worldwide for a number of years and increasing numbers of compounds continue to be detected in all kinds of waterbodies. Contaminated drinking water can be a major source for human uptake of organic micropollutants. Since groundwater is widely used for drinking water supplies worldwide it is of utmost importance to have a detailed understanding of the distribution and transport of these compounds in groundwater, in order to guarantee safe drinking water for mankind in the future. The World Health Organization's water safety plans also highlight the need for a profound understanding of the processes involved to protect groundwater as a drinking water resource (WHO, 2005).

Increasing numbers of different types of organic micropollutants have been identified and investigated in aqueous environments over recent decades, for example pharmaceutical compounds (Kümmerer, 2009; Li, 2014; Richardson and Bowron, 1985), pesticides (Ritter, 1990; Song et al., 2010; Yadav et al., 2015), and hormones (Kolpin et al., 2002; Silva et al., 2012; Young and Borch, 2012). A group of organic micropollutants that have recently come into focus comprises the perfluoroalkyl and polyfluoroalkyl substances (PFASs), which seem to be developing into a problem contaminant around the world (Kotthoff et al., 2015; Simon, 2014). Organic micropollutants have been detected in all parts of the hydrological cycle, including rainwater (Fernández-González et al., 2014; Guidotti et al., 2000), surface waters (e.g. Bu et al., 2015; Buser et al., 1998; Gros et al., 2007; Kolpin et al., 2002; Loos et al., 2009; Ternes, 1998), groundwater (e.g. Barnes et al., 2008; Halling-Sorensen et al., 1998; Lapworth et al., 2012; Loos et al., 2010), and drinking water (e.g. Heberer, 2002; Kunacheva et al., 2010; Leal et al., 2010; Post et al., 2012; Stackelberg et al., 2004; Stan and Heberer, 1997).

The main pathway into aquatic environments in rivers or lakes is through discharge from sewage treatment plants (Gros et al., 2007; Heberer et al., 2002; Jekel et al., 2015; Karthikeyan and Meyer, 2006; Metcalfe et al., 2003; Paxeus, 2004; Rabiet et al., 2006; Stackelberg et al., 2004). Organic micropollutants can also enter aquatic environments from livestock farming (Aga et al., 2003), landfill sites (Albaiges et al., 1986; Barnes et al., 2004; Holm et al., 1995), wastewater irrigation of fields (Scheytt et al., 1998; Ternes et al., 2007), leaking sewers (Fenz et al., 2005b; Gallert et al., 2005; Phillips et al., 2015), and on-site water treatment units and septic systems (Carrara et al., 2008; Godfrey et al., 2007). Although techniques exist (and are technically possible) for removing organic micropollutants during wastewater treatment (e.g. activated carbon adsorption, advanced oxidation processes, nanofiltration, reverse osmosis, and membrane bioreactors) standard wastewater treatment plants (WWTPs) do not usually completely remove all organic micropollutants (Evgenidou et al., 2015; Kreuzinger, 2008; Luo et al., 2014; Scheurer et al., 2010; Suárez et al., 2008; Tijani et al., 2013; Vona et al., 2015). This is largely due to the broad variety of organic compounds, each requiring different removal techniques due to their different chemical properties. Moreover, metabolites can form during WWTP processes (e.g. Boix et al., 2016; Evgenidou et al., 2015; Göbel et al., 2005).





Even irradiation with UV light (Bergheim et al., 2015) or ozonation (Favier et al., 2015) can lead to the formation of metabolites with higher ecotoxicity than their parent compounds. Accordingly, this implies that organic micropollutants are still continuously released into the aquatic environment and concentrations in aquatic environments are increasing rather than decreasing. This is a reason for great concern as both aquatic fauna and humans are, to a greater or lesser extent, exposed to

water that contains organic micropollutants. Adverse health effects on fish (e.g. reproductive and cytological effects) are frequently reported (Brooks, 2014; Overturf et al., 2015; Triebskorn et al., 2007). Numerous investigations have also been published into human health risks resulting from organic micropollutants in water (e.g. Rahman et al., 2009; Schwab et al., 2005; Stuart et al., 2012).

The number of investigations into organic micropollutants has increased in line with continuing improvements in analytical

techniques, such as the use of mass spectrometry (and enhancements) and solid phase extraction methods. These organic compounds can now be quantified down to low ng/l values (e.g. Nödler et al., 2010), which has opened up new possibilities for research into organic micropollutants including, for example, their use as anthropogenic indicators or tracers in aquatic systems. Albaiges et al. (1986) used caffeine as an indicator for groundwater pollution from a sanitary landfill. Seiler et al. (1999) investigated the potential of caffeine as a marker for anthropogenic nitrate in water wells, but found that its degradation

and sorption was too high for it to serve as an effective tracer. Fenz et al. (2005a) and Fenz et al. (2005b) demonstrated that the anti-epileptic drug carbamazepine is an excellent tracer of sewage exfiltration and can also be used to quantify sewage loss. Wolf et al. (2012) also used carbamazepine to detect sewage loss. However, although carbamazepine is useful for quantifying anthropogenic input or bank filtration (Clara et al., 2004; Kreuzinger et al., 2004; Massmann et al., 2008; Strauch et al., 2008), most pharmaceutical compounds can only be used as qualitative markers of anthropogenic impact. Other organic

micropollutants that have been reported to be good indicators of sewage leakage into groundwater are the X-ray contrast medium amidotrizoic acid (Wolf et al., 2004) and the artificial sweetener acesulfame (Wolf et al., 2012). Müller et al. (2012) used a time-dependent input function for different pharmaceuticals that took into account changes in prescriptions over the long term (either because of new pharmaceuticals becoming available or as a result of old compounds being made illegal) to identify the age and origin of groundwater at a former wastewater irrigation site. Nödler et al. (2013) used a combination of

micropollutants that had originated from wastewater (carbamazepine and acesulfame) or treated wastewater (valsartan acid) to determine the sources of groundwater in a karst spring (leaking sewer or rain-induced overflow of untreated sewage vs. outflow from a wastewater treatment plant). Zirlewagen et al. (2016) used artificial sweeteners to identify and quantify different sources as well as to estimate the residence times of wastewater contamination. Jekel et al. (2015) presented a schema for identifying processes and sources in the anthropogenic water cycle using different organic micropollutants that degrade

differently and have different transport properties. Nödler et al. (2012) identified two transformation products of sulfamethoxazole during batch experiments under varying redox conditions. Metabolization appeared to be partially reversible, with the concentrations of sulfamethoxazole increasing and concentrations of transformation products decreasing following the complete consumption of nitrate. Since the same transformation products were found in a karstic spring this relationship may be useful as an indicator of nitrification and denitrification processes in aqueous environments. Since many organic



micropollutants show redox-sensitive behavior under groundwater conditions (e.g. Banzhaf et al., 2012; Barbieri et al., 2011; Burke et al., 2014; Heberer et al., 2008; Massmann et al., 2008) it may be possible to use such compounds or groups of compounds to determine redox zones in complex anthropogenically-influenced settings. Greskowiak et al. (2006) were able to demonstrate a correlation between temperature-dependent redox zonation in groundwater and the fate of phenazone at an

infiltration site in Berlin.

In view of the above-mentioned concerns regarding the presence of organic micropollutants in aquatic environments, and especially in groundwater, there is a clear need to develop a sound understanding of how they are transported and behave in groundwater. Laboratory experiments on the transport and eventual fate of organic micropollutants under defined boundary conditions will always be important because the boundary conditions for field studies are poorly known, which affects the

transferability of their results to other systems. This paper therefore provides a review of published column experiments investigating the properties and transport behavior of organic micropollutants, since such experiments provide a suitable setup for this task. The relevant transport properties of organic micropollutants are first presented, followed by a discussion of which compounds and which of their properties can be investigated using the experimental setup of a column experiment. The weaknesses and problems, as well as the advantages, of different experimental setups will be discussed and finally, other

laboratory methods that can be used to investigate organic micropollutants are compared with column experiments.

## 2 Factors affecting the transport of micropollutants in groundwater and in column experiments

The transport of (organic) solutes in groundwater depends on the chemical and physical properties of the compounds, the solvent (i.e. the groundwater, including all solutes), and the substrate (the aquifer material). The main processes of solute transport are advection and hydrodynamic dispersion. The movement of solutes can be retarded compared to that of the

containing groundwater, mainly as a result of sorption. Oxidation-reduction reactions, precipitation-dissolution reactions, and mechanical filtering are other mechanisms that can also reduce the velocity of solutes during their transport in groundwater, which is only driven by advection and hydrodynamic dispersion. Substances that behave in a very similar way to groundwater are known as conservative tracers. Such substances (e.g. bromide, chloride, uranine, eosine, etc.) are used, not only in laboratory experiments but also in field tracer tests, to identify differences between the transport behavior of reactive

substances and the groundwater movement.

The effect that different factors have on solute transport in groundwater is shown as a schematic breakthrough curve in Fig. 1. A conservative tracer or substance does not react with the soil and/or groundwater, nor does it undergo biological or radioactive decay (Fetter, 1988). It is only influenced by advection and hydrodynamic dispersion (the red breakthrough curve in Fig. 1). In contrast, a non-conservative tracer or substance is likely to react with the soil and/or groundwater and its transport will be

influenced by many (or all) of the controlling factors for solute transport described below.





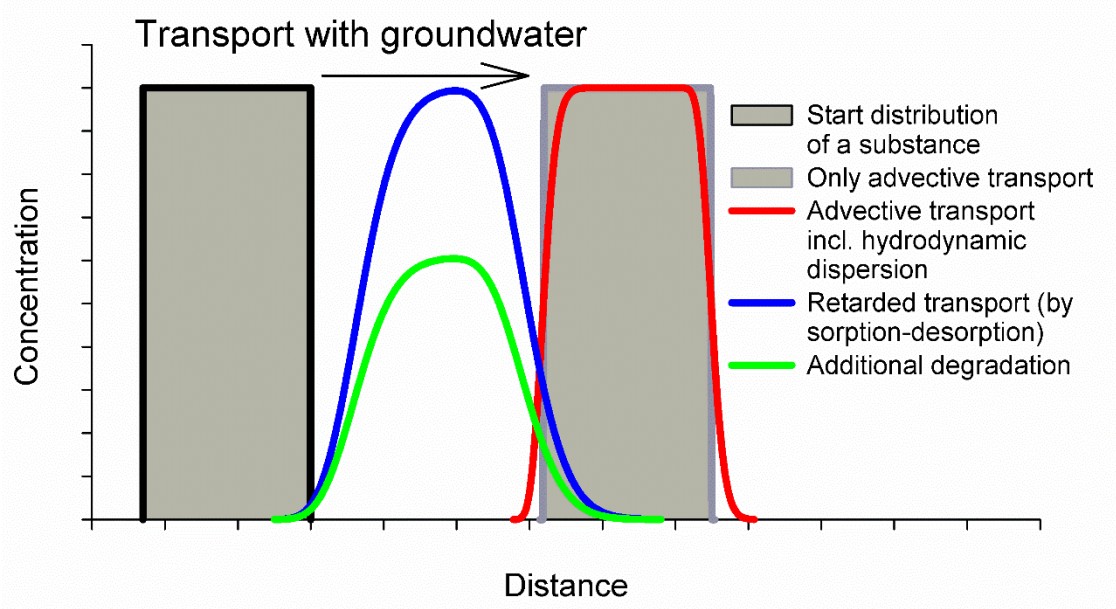

Figure 1: Schematic representation of solute transport in groundwater, taking into account the main transport processes of advection, hydrodynamic dispersion, retardation, and degradation.

## 2.1 Advection

The most important factor influencing solute transport in groundwater or during a column experiment is advection, which involves the transport of dissolved or suspended matter with water. In Fig. 1 the gray rectangle with black solid borders represents the start distribution of a given substance or compound and its area represents the mass of the substance or compound (i.e. its concentration times the distance). If advection is the only factor affecting the transport of this substance within the groundwater or column, the rectangle will move along the flow path without any alteration (as indicated by the gray rectangle

with gray borders on the right hand side of Fig. 1).

## 2.2 Hydrodynamic dispersion: diffusion and mechanical dispersion

Diffusion results in the equalization of concentration differences, following Fick's law. Diffusion is stronger at low flow velocities and small scales and becomes almost insignificant as the velocities and scale increase. The impact of diffusion within an aquifer is therefore often negligible. However, diffusion can be significant if a column experiment is operated at low flow

velocities or at a very small scale.

Mechanical dispersion results in a spreading of the concentration front of a solute during transport and is caused by velocity and flow path variations within an aquifer due to inhomogeneities on both micro and macro scales (Domenico and Schwartz, 1998). It results in both transverse (or lateral) dispersion (i.e. a spreading of the solute perpendicular to the flow direction) and





longitudinal dispersion (differences in travel time, Appelo and Postma, 2005). The impact of dispersion on mass transport in groundwater increases with velocity and scale.

The combination of diffusion and mechanical dispersion is known as hydrodynamic dispersion; it leads to a broadening of the breakthrough curve at a particular observation point during flow through a porous medium (the red curve in Fig. 1).

## 2.3 Retardation

Retardation is the delay in the transport of a substance relative to the groundwater flow. There are many possible reasons for the retardation of a reactive solute but the main mechanism involved is sorption, which involves reactions between solutes and solid surfaces (Domenico and Schwartz, 1998). Sorption can occur either by adsorption onto surfaces or by absorption into the substrate. The opposite process is desorption. Sorption and desorption of various species often occur as coupled processes in which dissolved ions of one species displace sorbed ions of another species from their sorption sites in the substrate. These processes are dependent on combinations of various specific properties of the compounds, such as their charge, the size of the ions, their concentration in the solution, and on the availability of sorption sites. Combined sorption-desorption processes are known as ion exchange. The increase in solute concentration during a breakthrough typically leads to changes in the equilibrium between the solid and liquid phases of the compound. A compound can only break through when all sorption places are filled according to the new equilibrium. When the system is flushed with compound-free water the opposite process takes place, the equilibrium shifts back, and the sorbed compounds are again released into the solution. The result is a delayed breakthrough curve at the observation point (blue curve in Fig. 1). Typical retardation values range between 1.2 and 10 (Langguth and Voigt, 2004). Since it is difficult to observe sorption-desorption processes directly, either in the field or in a laboratory, the retardation of a breakthrough curve is commonly used as an indicator for these processes.

Schaffer and Licha (2015) described the two main sorption processes as (i) hydrophobic sorption, and (ii) ionic sorption. Hydrophobic sorption occurs during the transport of non-polar, non-ionized (neutral) compounds, which sorb onto the uncharged sites of organic matter. Neutral compounds often show a degree of hydrophobic behavior due to the polarity of the water molecules and they therefore have a tendency to accumulate in non-polar environments (solid or liquid organic phases). Organic matter is one of the best sorbents for non-polar organic compounds because of its large surfaces (Delle Site, 2001). If a soil contains more than 0.1% organic carbon the adsorption of nonionic organic substances is attributed entirely to the organic carbon (Appelo and Postma, 2005). Ionic sorption occurs during the transport of polar, ionized (anionic or cationic) compounds. Non-neutral compounds often show more hydrophilic behavior and are therefore often more conservative than neutral compounds. However, in the presence of charged sorption sites (clay minerals, organic matter, Fe/Mg-(hydro)-oxides) these compounds can interact in many ways with the substrate through, for example, ion exchange, surface complexation, hydrogen bonding, ligand exchange, cation bridging, or charge-transfer processes (Schaffer and Licha, 2015). The behavior of organic compounds during transport in groundwater is therefore clearly dependent on their speciation (neutral, anionic/acid, cationic/basic, anionic-cationic/zwitter-ionic). The speciation of organic compounds depends on (a) their molecular structure, and (b) the pH of the water. Many (but not all) organic molecules possess one or more functional groups from which protons





can dissociate. The charge of such molecules can therefore change from positive (cationic) to neutral or negative (anionic) depending on the degree of dissociation. It is even possible for molecules to bear differently charged (positive and negative) functional groups at the same time and to then behave as both anions and cations (zwitter ions).

The species of dissociative organic compounds in groundwater is controlled by the dissociation constant of organic acids, $K_a$, and (if present) organic bases, $K_b$ (Schaffer and Licha, 2014). The relationship between the inverse common logarithms of the acidic/basic dissociation constants $pK_a$ and $pK_b$ (i.e. the inverse common logarithm of the concentration of dissociated protons) and the pH defines the charge state of the compound. Where the pH $<<$ $pK_a$ (which is generally the case for $pK_a > 10$ at normal groundwater pH values of between 6 and 8) the substance will exist in its undissociated (neutral) form, and where the pH $>>$ $pK_a$ (which is generally the case for $pK_a$ values between 0 and 3 at normal groundwater pH values) the substance will exist in its dissociated (anionic/acidic) form. The relationship between $pK_b$ and pH is the opposite of that between $pK_a$ and pH: where the pH $<<$ $pK_b$ the substance will exist in its dissociated (cationic/basic) form and where the pH $>>$ $pK_b$ it will exist in its undissociated (neutral) form. Substances with $pK_a$ or $pK_b$ values close to the pH of the solvent will exist in both non-polar and polar forms.

The pH of groundwater therefore defines the polar character of organic compounds and consequently has an important influence on the sorption behavior of the compounds (Schwarzenbach et al., 2003).

The distribution coefficient ($K_{OW}$ or P) between water and octanol is used to describe the hydrophilic or hydrophobic character of a non-ionizable or neutral organic compound. The distribution coefficient for ionizable or non-neutral compounds is D, which is pH-dependent.

$$K_{OW} = P = D_{pH-dependent} = \frac{c_{n-octanol}}{c_{water}} \qquad (1)$$

A compound with a distribution coefficient of $\log K_{OW}$ (= log P or log D) $> 0$ is more lipophilic (i.e. migrates into the organic phase), and $< 0$ is more hydrophilic (i.e. migrates into the aqueous phase). Predicted values for P and D at specific pH values can be obtained from databases (e.g. Chemicalize.org; SciFinder) or from chemical drawing programs (e.g. ChemDraw; MarvinSketch; Smyx Draw).

In order to obtain the distribution coefficient between a solid and liquid phase ($K_d$) the $K_{OC}$ must be corrected because $\log K_{OC}$ describes the partitioning between water and a 100% organic carbon phase:

$$K_d = K_{OC} \cdot f_{OC} \qquad (2)$$

with $f_{OC}$ as the fraction of organic carbon (Appelo and Postma, 2005).

If the distribution coefficient between a solid and liquid phase ($K_d$) of a specific compound is known, together with the bulk density ρ and the porosity Θ of the substrate, the retardation factor of this compound can be approximated as follows (Stumm and Morgan, 1996):

$$R_f \approx 1 + \frac{\rho}{\theta} \cdot K_d \qquad (3)$$



## 2.4 Degradation

Degradation is the mineralization of complex molecules to form inorganic molecules or elements. A reactive substance can undergo chemical, biological, or radioactive degradation during transport, which will reduce both the concentration and the total content of that particular substance (Fetter, 1988 and Fig. 1), and hence lower the breakthrough curve (green curve in Fig.

1). Radioactive decay is not relevant for organic compounds but both microbial degradation and chemical degradation (e.g. hydrolysis, oxidation-reduction reactions, UV degradation) are potentially significant. Redox reactions are electron transfer reactions that induce metabolism of organic molecules. This can lead to total mineralization and to the formation of metabolites with properties that may differ from those of the original molecules. In most cases these processes are catalyzed by microbes (Schwarzenbach et al., 2003). Redox reactions during groundwater transport as a result of changes in thermodynamic

conditions along the flow path can also affect the solubility of a compound. Redox processes control the natural concentrations of $O_2$, $Fe^{2+}$/$Fe^{3+}$, $SO_4^{2-}$, $H_2S$, $CH_4$, and other compounds in groundwater (Appelo and Postma, 2005), and can therefore induce zones with different oxygen concentrations, zones of nitrate-reduction, iron-reduction, sulfate-reduction, or methanogenesis, and zones with different pH values. This chemical zoning may then produce precipitation-dissolution reactions in the transported compounds, which can result in mass loss for a particular substance during transport and therefore

appear as degradation.

## 2.5 Transport behavior of organic micropollutants

Organic compounds include a very wide range of the compound-specific properties described above. Different species and metabolites of a compound can exist in the same sample. Transport behavior is furthermore dependent on the pH of the fluid and on the properties of the substrate (e.g. the organic carbon content, the presence of sorption sites on clay minerals, or the

presence of Fe/Mg oxides or hydroxides). It is therefore not possible to describe the general behavior of individual compounds or groups of compounds in groundwater. It is even difficult to predict the behavior of particular compounds in specific settings because of the large number of defining variables. The future objective should therefore be to progress from individual case studies to a general process understanding, with the ultimate aim being to be able to make general predictions concerning the behavior and eventual fate of organic compounds in natural aqueous environments.

## 3 Column experiments on organic micropollutants

### 3.1 Setup and evaluation of laboratory column experiments

Laboratory column experiments can be used for many different applications. The boundary conditions and experimental setup can be varied to best address particular research questions or compounds. Column experiments are generally used to investigate the transport behavior, sorption, and degradation of a specific compound, or group of substances. However, more specific

issues can also be addressed such as the effect of entrapped air on soil permeability (Christiansen, 1944), the effect of a



fluctuating water table on redox conditions (Sinke et al., 1998), the effect of preferential flow on solute transport (Schoen et al., 1999), the effect of a sterilized soil on sorption (Lotrario et al., 1995), or the influence of methanol on the retention of hydrophobic organic chemicals (Nay et al., 1999). An overview of the types of investigations that can be carried out with column experiments is provided below, the methods involved presented, and practical issues discussed. Additional information

on column experiments for specific compounds is provided in Section 3.2.

### 3.1.1 Technical setup

The basic principle of a column experiment is to pump water with a specific composition, including solutes of interest, through a column filled with a specific substrate. Since the aim of column experiments is to investigate (dynamic) processes it is common to start with a "neutral" fluid that is in hydrochemical equilibrium with the substrate, and then switch to the actual

test fluid containing the solutes of interest and a mandatory conservative tracer. Figure 2 shows three basic setups for laboratory column experiments. The normal practice is to operate columns in an upright position. This allows percolation through the unsaturated zone to be simulated, as shown in Fig. 2(a). Different degrees of saturation can thus be achieved within the column and the effects of a fluctuating water table can also be simulated. In order to simulate saturated groundwater conditions the column can be operated with an upward flow direction, as shown in Fig. 2(b). This results in saturated conditions throughout

the column and may help to avoid any entrapment of gas bubbles. Different parameters or boundary conditions and their evolution along a flow path can be investigated by coupling several columns together, as shown in Fig. 2(c).





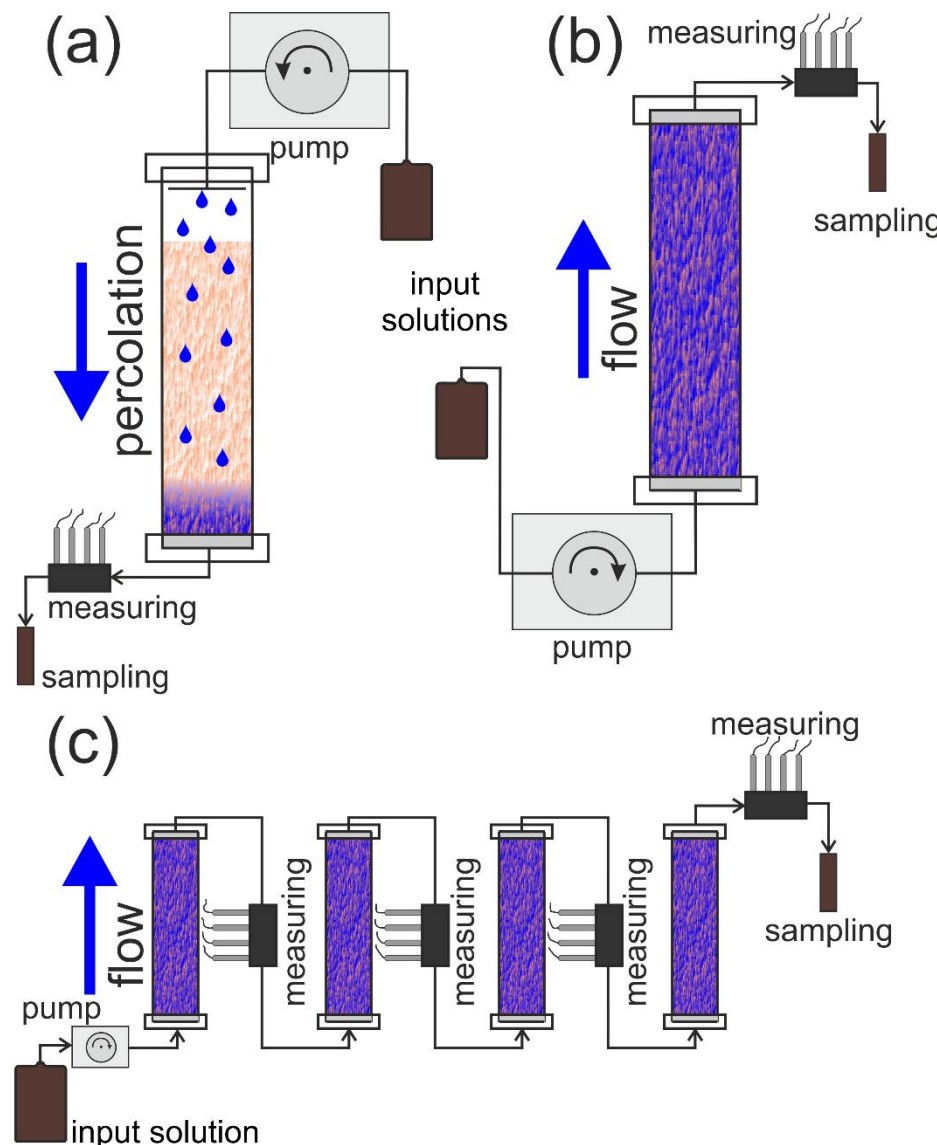

Figure 2: Schema of common setups used in columns experiments: (a) unsaturated / leaching experiments (e.g. Siemens et al., 2010); * a fluctuating water table is established in some experimental setups (e.g. Sinke et al., 1998), (b) saturated / groundwater flow experiments (e.g. Scheytt et al., 2004), (c) saturated columns connected in series in order to measure the evolution of hydrochemical parameters (such as the oxidation-reduction potential) along the flow path (e.g. Nay et al., 1999).

Columns are typically made of stainless steel (Alotaibi et al., 2015; Banzhaf et al., 2012; Burke et al., 2013; Schaffer et al., 2015; Unold et al., 2009; Xu et al., 2010) in order to prevent interactions with solutes, or from acrylic glass (Gruenheid et al., 2008; Hebig et al., submitted; Rauch-Williams et al., 2010; Yao et al., 2012) for improved visual control of saturation levels and tracer transport (where a dye tracer is used). Although acrylic glass is a synthetic plastic it is reported to be inert to many



organic solutes (Hebig et al., 2014) and therefore suitable for column experiments with organic micropollutants. Glass columns are also often used (e.g. Estrella et al., 1993; Fan et al., 2011; Nay et al., 1999; Persson et al., 2008; Simon et al., 2000) since glass is assumed to be inert with respect to organic compounds. Other materials used for columns include PVC (e.g. Bertelkamp et al., 2012; Greenhagen et al., 2014; Salvia et al., 2014; Sinke et al., 1998), polyethylene (Bertelkamp et al., 2012), and

aluminum (Burke et al., 2014).

The physical dimensions of columns are not yet standardized. Short columns enable fast experiments and many repetitions, while longer columns need more time for equilibration before the actual start of the experiment but allow longer reaction times (which means that more reactions can be distinguished) and more complex settings (e.g. the establishment of redox zonings). The dimensions of columns vary considerably due to the broad range of research applications; for example, lengths have been

reported ranging from 5 cm (Estrella et al., 1993; Teijón et al., 2014) to 2.4 m (Cordy et al., 2004), and inner diameters have been reported ranging from 2 cm (Teijón et al., 2014) to 36 cm (Bertelkamp et al., 2012; Bertelkamp et al., 2014). The choice of a reasonable length-to-diameter ratio is important if scaling effects are to be avoided. In long columns with small diameters the main flow (and hence the main transport) can occur by preferential flow along the boundary between the inner column surface and the sediment grains. If the diameter is too large transversal dispersivity can become significant for solute transport,

which makes analysis even more complex (column experiments are generally designed to exclude transversal dispersivity from the transport model). Using a short column with too large a diameter relative to its length may prevent uniform, homogeneous flow within the sediment. Lewis and Sjöström (2010), in their extensive review of the design of column experiments, recommended a diameter-to-length ratio of 1:4 in order to avoid such effects. However, this ratio is rarely reported in published literature.

The tubing and other materials reported are commonly made from Teflon/PTFE (e.g. De Wilde et al., 2009; Fan et al., 2011; Ke et al., 2012; Strauss et al., 2011; Teijón et al., 2014) or stainless steel (e.g. Ke et al., 2012; Nay et al., 1999; Teijón et al., 2014), which are known to be inert materials. Other materials that have been used in column experiments are Pharmed tubing (Strauss et al., 2011), (dark) polyethylene (Bertelkamp et al., 2012; Bertelkamp et al., 2014), PVC, (black) Tygon tubing, vinyl, polysulfone, brass (Greenhagen et al., 2014), polypropylene, and silicone (Srivastava et al., 2009). However, the influence of

many of these reported materials on organic compounds is unknown and may be problematic, as is the case for Pharmed tubing, silicone, and Tygon tubing (Hebig et al., 2014). Unfortunately, there is often little information provided concerning the materials used (in particular the filter materials used), even though they can have a significant influence on the mass recoveries of organic micropollutants. Only a few investigations have specifically addressed the issue of possible interactions between the materials used in experiments and the compounds and fluids under investigation, or included preliminary investigations to

allow such interactions to be avoided (Greenhagen et al., 2014; Gruenheid et al., 2008; Hebig et al., 2014; Srivastava et al., 2009). This often untested impact of laboratory materials on compound concentrations may therefore have a significant (but unknown) influence on the results of column experiments.

Peristaltic pumps are commonly used in investigations into fluid transport in saturated columns (e.g. Banzhaf et al., 2012; Bertelkamp et al., 2014; De Wilde et al., 2009; Müller et al., 2013; Rodriguez-Cruz et al., 2007) because they are able to





provide uniform flow, even at low flow rates. Other pumps that have been used include pulsating pumps (Massmann et al., 2008), piston pumps (Persson et al., 2008), gear pumps (Alotaibi et al., 2015) and, in unsaturated and leaching experiments, suction pumps (Salem Attia et al., 2013; Siemens et al., 2010). In leaching experiments it is also common to simply let the fluid leach under gravity (Scheytt et al., 2006; Scheytt et al., 2007; Xu et al., 2010).

To prevent inhomogeneous flow and mass transport through the column a layer of clean, well sorted filter quartz sand is often included at both the inlet and the outlet ends of the column (e.g. Banzhaf et al., 2012; Gruenheid et al., 2008; Persson et al., 2008; Unold et al., 2009). Other materials reported to have been used as filters are glass beads or globes (e.g. Salvia et al., 2014; Scheytt et al., 2004; Scheytt et al., 2006), and glass wool (Nay et al., 1999; Salem Attia et al., 2013). These filter layers are often combined with manufactured filters such as perforated PVC (Bertelkamp et al., 2014) or stainless steel plates (Strauss
et al., 2011), stainless steel meshes (Burke et al., 2013; Fan et al., 2011; Patterson et al., 2010; Yao et al., 2012), glass fiber filters (Strauss et al., 2011), cheesecloths (Fan et al., 2011; Srivastava et al., 2009), aluminum screens (Greenhagen et al., 2014), porous HDPE (Lorphensri et al., 2007), Teflon gauze nets (Scheytt et al., 2004, personal communication), porous glass (Siemens et al., 2010), porous ceramic plates (Unold et al., 2009), or paper filters (Srivastava et al., 2009). In this way the hydraulic contrast between the tube/inlet and the substrate can be reduced and the incoming water and solutes spread over the
entire width of the column. Any intrusion or wash-out of finer sand particles through the inlet or outlet of the column should also be avoided through the use of such filters.

A wide range of substrates have been used as filling for columns in published research, depending on the specific objectives. These include natural (site-specific) aquifer sediment (e.g. Alotaibi et al., 2015; Burke et al., 2013; Lopez-Blanco et al., 2005; Mersmann et al., 2002; Preuss et al., 2001; Teijón et al., 2014), natural soil (e.g. Aga et al., 2003; Cordy et al., 2004; Kamra
et al., 2001; Murillo-Torres et al., 2012; Rodriguez-Cruz et al., 2007; Xu et al., 2010), artificial soil (De Wilde et al., 2009), artificial (model) sediments such as well sorted filter sands or technical quartz sands (e.g. Baumgarten et al., 2011; Bertelkamp et al., 2014; Greenhagen et al., 2014; Nay et al., 1999), and other artificial materials such as iron coated sand (Hebig et al., submitted), magnetic nanoparticle-coated zeolite (Salem Attia et al., 2013), alumina, silica gel (Lorphensri et al., 2007), and biochar (Yao et al., 2012). The substrate can be installed wet or saturated (e.g. Alotaibi et al., 2015; Nay et al., 1999; Simon et
al., 2000), or dry (e.g. Banzhaf et al., 2012; Fan et al., 2011; Scheytt et al., 2007; Sinke et al., 1998; Teijón et al., 2014), ideally in small (1-2 cm) layers that are individually compacted using a tool such as a stamp (Banzhaf et al., 2012), plunger (Scheytt et al., 2007), or pestle (Unold et al., 2009), or alternatively by vibration (Burke et al., 2013; Teijón et al., 2014), by tapping against the column (Bertelkamp et al., 2014; Strauss et al., 2011), or by placing a weight on top before installing the next layer (De Wilde et al., 2009). Some experiments have been performed using undisturbed soils or sediments (e.g. Greenhagen et al.,
2014; Massmann et al., 2008; Muñoz-Leoz et al., 2011). However, even when great care is taken over filling the column the resulting effective porosities are usually higher than in natural sediments. The reported range of effective porosities is between 0.28 (Greenhagen et al., 2014) and 0.49 (Schaffer et al., 2015), of which only the lower limit is representative of effective porosities found in natural aquifers. It appears that effective porosities lower than 0.30 are only achieved when the column is filled with undisturbed sediments. The high effective porosities in most experiments may lead to lower flow velocities and



lower reactive surface areas than would be expected in a natural environment and may therefore be responsible for the often noted differences between laboratory results and results from field tests. However, many published investigations either do not specify the effective porosities or only report the total porosities (from the ratio of the weight of the column filled with dry sediment to the weight of the column with fully saturated sediment). Pore water velocities vary according to the experimental

conditions. The flow velocity should ideally reproduce natural groundwater flow velocities, which one would normally expect to be between 1 cm d$^{-1}$ and 1 m d$^{-1}$. Using higher velocities allows experiments to be completed more quickly and hence many repetitions, but slow velocities are more likely to provide a realistic representation of natural processes, involving equilibration of solute and solid phases. Flow velocities in columns experiments are typically between 4 cm d$^{-1}$ (Strauss et al., 2011) and 348 cm d$^{-1}$ (Fan et al., 2011). There are similar uncertainties in reported flow velocities to those previously mentioned for

porosities, as authors do not always state clearly whether they are referring to pore water velocity or Darcy velocity. Various types of fluid have been used as solvents in column experiments depending on their objectives, including natural (site-specific) groundwater (Greenhagen et al., 2014; Mersmann et al., 2002; Preuss et al., 2001; Scheytt et al., 2004) or surface water (e.g. Banzhaf et al., 2012; Baumgarten et al., 2011; Burke et al., 2014; Schaffer et al., 2012), model or artificial (ground)water (De Wilde et al., 2009; Estrella et al., 1993; Murillo-Torres et al., 2012; Xu et al., 2010), treated wastewater (e.g. Alotaibi et al.,

2015; Cordy et al., 2004; Ke et al., 2012), tap water (Bertelkamp et al., 2012; Burke et al., 2013), distilled water (Aga et al., 2003), ultrapure water (Simon et al., 2000), and even liquid manure (Strauss et al., 2011).

The reported solute concentrations vary considerably depending on the objectives of the individual experiments, ranging between 60 ng L$^{-1}$ (Rauch-Williams et al., 2010) and 2 mg L$^{-1}$ (Siemens et al., 2010; Yao et al., 2012). Some investigations therefore use concentrations that are found in natural environments or in waste waters, while others may aim to identify specific

processes and therefore use higher concentrations to enhance the effect of the processes. Higher concentrations are also easier to measure.

An important basic parameter is the pH of the fluid used in column experiments as the polar character of many organic micropollutants varies according to the relationship between the pKa value and the pH. An organic compound may therefore be persistent and highly mobile under the specific conditions of one experiment but be strongly retarded in another experiment,

due to the different pH values of the fluid used.

Field parameters and/or tracers are commonly measured using flowthrough cells fitted with probes (e.g. Mersmann et al., 2002; Müller et al., 2013). Sampling for solutes can be performed in a number of different ways including "by hand" (bottle, beaker, e.g. Burke et al., 2014; Cordy et al., 2004; Hebig et al., submitted), using an automated fraction collector (e.g. Banzhaf et al., 2012; Rodriguez-Cruz et al., 2007; Scheytt et al., 2004; Srivastava et al., 2009; Unold et al., 2009), using sampling ports

attached laterally to the column (Alotaibi et al., 2015; Baumgarten et al., 2011; Burke et al., 2014), or online and in real-time using a spectrometer at the outlet from the column (Teijón et al., 2014). Sampling ports alongside the column risk altering the hydraulics within the column and should only be used with great caution, using small volumes and low sampling rates. A further means of assessing the fate of organic compounds in a column experiment is to extract the substrate from the column after completing the experiment and to analyze the solid phase for irreversibly sorbed solutes (Banzhaf et al., 2012). This



enables all parts of the mass balance to be determined since the mass of solute degraded can be determined by deducting the recovered mass and the irreversibly sorbed mass from the injected mass.

### 3.1.2 Significance and influence of boundary conditions and heterogeneity

The boundary conditions of column experiments need to be known in order to ensure correct interpretation of the results obtained. Van Genuchten and Parker (1984) discussed the physical and mathematical significance of the boundary conditions that apply to one-dimensional solute transport in laboratory column experiments. They presented solutions of the convective-dispersive transport equation that can be used to analyze column effluent data. Leij et al. (1993) investigated analytical solutions for non-equilibrium solute transport in 3-dimensional porous media. They found that the effect of non-equilibrium on 1D transport is similar to that on 3D transport. The effect of non-uniform boundary conditions on steady flow in saturated homogeneous cylindrical soil columns was investigated by Barry (2009). He reported that uniform flow in the column could be achieved if a baffle zone was established at each end of the column. Sentenac et al. (2001) used fiber-optic sensing to measure side-wall boundary effects in soil columns. They detected flow velocity differences between the center of the column and the boundary wall, which surface roughness was varied. Seyfried and Rao (1987) described preferential flow effects in columns containing undisturbed substrates. They assumed that flow occurred through series of large pores or pore sequences. Perret et al. (2000) investigated preferential solute flow in columns containing undisturbed substrates using a tomography technique derived from medical applications. This technique allowed real-time analysis of the flow patterns of radioactive tracers in both 2D and 3D.

The dimensions of a column used for in-column experiments appear to have a marked influence on the hydraulic conditions within the column and on the transport of the investigated compounds (see Sections 2 and 3.1.1). Wierenga and Van Genuchten (1989) found that the dispersivity of chloride and tritium tracers used in unsaturated column experiments increased significantly with column length, while retardation factors remained essentially the same. However, Ribeiro et al. (2011) showed that the retardation factor for potassium ions investigated during leaching experiments decreased with increasing column length, and that the dispersive-diffusive coefficient and dispersivity both increased with increasing pore-water velocity and increasing soil column length. Rühle et al. (2013) described changes over time to the water flow through the porous medium in a column, from uniform to non-uniform. They concluded that flow path changes occurred due to clogging of small pores near the column inlet as a result of microbial growth and calcite precipitation, which then caused non-uniform water flow and solute transport. Bromly et al. (2007) reviewed published data on experiments on almost 300 repacked saturated homogeneous column experiments. They related the dispersivity to the length of the column used, i.e. short columns had greater dispersivities. However, clay content was identified as the most important factor controlling dispersivity, followed by the diameter of the column. However, they pointed out that the individual experimental design (e.g. the column geometry, inlet dead volume, and soil packing) needs to be taken into account in order to be able to relate dispersivity to soil properties. Although not specifically addressing column experiments, Schulze-Makuch (2005) provided a useful overview of longitudinal dispersivity data for different materials and implications for scaling behavior. Nimmo and Akstin (1988) investigated the





hydraulic conductivity of a sandy soil with a low water content following compaction by various methods and found that conductivities varied by almost three orders of magnitude depending on the compaction.

### 3.1.3 Evaluation of column experiments

It should be pointed out that the main issues when evaluating and interpreting results from column experiments are the same for different organic compounds, these being the interactions with surfaces and the ionic or neutral character of the molecules. In order to describe the transport behavior of organic solutes in column experiments their (reactive) breakthrough curves are often compared with the breakthrough curves of a conservative tracer (which represent the flow velocity of the fluid). This is sometimes performed by graphical analysis (e.g. Scheytt et al., 2004). In both reactive and conservative breakthrough curves the instant is identified at which the observed concentration reaches 50% of the injected concentration (i.e. c/c0 = 0.5). The temporal offset between those two points in time represents the retardation of the reactive compound compared to the tracer or fluid. In this approach the mass recovery (as a measure of the degradation) is calculated from the area below the breakthrough curve and the pumped volume.

Time (or temporal) moment analysis has also frequently been used to evaluate column experiments (Kamra et al., 2001; Murillo-Torres et al., 2012). This method allows the mean breakthrough time, spreading, and asymmetry of a breakthrough curve to be characterized by integration of the breakthrough curve (Appelo and Postma, 2005; Hebig et al., 2015). It is, however, more common to model (or "fit") the solute concentrations to the convective-dispersive equation (CDE):

$$R_f \frac{\partial c_r}{\partial t} = D \frac{\partial^2 c_r}{\partial x^2} - v \frac{\partial c_r}{\partial x} - \mu c_r \tag{4}$$

in which the term on the left-hand side describes the retardation of a solute, and the terms on the right-hand side describe the dispersion, the average pore-water velocity, and the first-order decay (i.e. degradation) of the solute, (described in detail in, e.g. Parker and Vangenuchten, 1984; Toride et al., 1999). Since the retardation of a conservative tracer is defined as 1 (i.e. no retardation compared to the pore-water velocity) and both the pore-water velocity and the dispersion are aquifer-specific parameters, the breakthrough curve of a conservative tracer can be used to iteratively vary both parameters until the "fit" matches the observed concentrations sufficiently well. Knowing the dispersion and velocity in the experiment makes it possible to reduce the number of variables for the reactive compounds and to then repeat the fitting procedure in order to estimate the retardation and degradation.

Computer software commonly used to fit the CDE to observed breakthrough curves is the CXTFIT software (Toride et al., 1999) which is available in various forms, for example through the public domain STANMOD software (Šimůnek et al., 1999), and can be downloaded at no cost (e.g. Bertelkamp et al., 2014; Fan et al., 2011; Kamra et al., 2001; Müller et al., 2013; Persson et al., 2008; Schaffer et al., 2015; Unold et al., 2009). The free HYDRUS-1D software (Šimůnek et al., 2009) is also frequently used for transport parameter determination (De Wilde et al., 2009; Strauss et al., 2011; Teijón et al., 2014). Other relevant available software packages are Origin (Microcal, 1995), as used for example by Alotaibi et al. (2015), PHREEQC



**3.2 Column experiments on different groups of organic micropollutants**

Column experiments on different groups of organic micropollutants are described in greater detail below, distinguishing
between investigations into pharmaceuticals, pesticides, and other organic micropollutants. Selected column experiments for
non-organic compounds are also presented for comparison. The presented examples of column experiment are then
summarized and discussed.

**3.2.1 Pharmaceuticals**

Column experiments are often used to investigate the transport of pharmaceutical compounds under both saturated and
unsaturated conditions. These are presented separately in this section, beginning with experiments under saturated conditions.
Mersmann et al. (2002) investigated the transport of carbamazepine, clofibric acid, diclofenac, ibuprofen, and propyphenazone
under saturated conditions and found the transport of all of these compounds to be unaffected by changes in the pH,
temperature, dissolved oxygen content, or ion concentration of the water used. Moreover, they found that each of these
compounds except for clofibric acid was retarded in the column. In contrast, Gruenheid et al. (2008) found that temperature
affected the degradation of sulfamethoxazole under saturated conditions, with higher temperatures resulting in increased
mineralization. However, the biodegradation of iopromide was high at all investigated temperatures (5, 15, and 25°C) while
the naphthalenedisulfonic acids showed no biodegradation at any of these temperatures. Müller et al. (2013) investigated the
influence of different total organic carbon contents and the resulting changes in redox conditions on the transport of primidone,
carbamazepine, and sulfamethoxazole. Carbamazepine and primidone were found to be retarded in the presence of organic
matter while sulfamethoxazole remained unaffected. However, none of the three compounds was degraded under any of the
investigated conditions. Banzhaf et al. (2012) also carried out saturated column experiments under varying redox conditions.
They found that the degradation of sulfamethoxazole was clearly redox dependent, exhibiting strong degradation associated
with denitrification. Both retardation and degradation were observed for carbamazepine. Retardation in the column was
observed for diclofenac but not for ibuprofen. Scheytt et al. (2004) performed experiments under saturated conditions and
found neither degradation nor retardation for clofibric acid, whereas diclofenac and propyphenazone were both retarded.
Patterson et al. (2010) observed no rapid degradation of carbamazepine and oxazepam under saturated experimental conditions.
Teijón et al. (2014) investigated naproxen and found that the sorption of this compound was independent of the flow rate
during the experiment; they noted a generally low sorption affinity for naproxen. Bertelkamp et al. (2014) investigated sorption
and biodegradation for ibuprofen, ketoprofen, gemfibrozil, acetaminophen, trimethoprim, propranolol, metoprolol,
carbamazepine, and phenytoin under oxic conditions, in order to simulate bank filtration. They found that the biodegradation
of these compounds could be predicted on the basis of their functional groups. Moreover, they observed no retardation for any
of the investigated compounds. Hebig et al. (submitted) showed a strong correlation between retardation and organic carbon




content for the anionic forms of gemfibrozil and ibuprofen, and also for the neutral carbamazepine, all of which were lipophilic (log D > 0). The neutral compounds sulfamethoxazole and caffeine, both of which were hydrophilic under the pH of the experiments, were strongly degraded in the presence of organic matter.

Unold et al. (2009) investigated sulfadiazine under near saturated conditions and found the degradation to be light dependent. Furthermore, sulfadiazine was sorbed onto the upper layer of the investigated soil column, despite showing a high leaching potential. Oppel et al. (2004) investigated the leaching of pharmaceuticals in unsaturated soil columns. They found a low leaching potential for diazepam, ibuprofen, ivermectin, and carbamazepine, while clofibric acid and iopromide were highly mobile. Siemens et al. (2010) also carried out leaching experiments under unsaturated conditions with naproxen, ibuprofen, bezafibrate, diclofenac, gemfibrozil, clarithromycin, trimethoprim, clindamycin, erythromycin, and metoprolol; they found that the investigated clay soil had significant potential to retain these pharmaceuticals. The retention capacity was, however, limited and all compounds were leached to some extent. Wu et al. (2010) found low mobility for carbamazepine, diphenhydramine, fluoxetine, diltiazem, and clindamycin, and also for two metabolites (carbamazepine-10,11-epoxide and norfluoxetine), in unsaturated leaching experiments. Moreover, carbamazepine, diphenhydramine, and fluoxetine were persistent throughout the experiment. Leaching experiments carried out by Salvia et al. (2014) indicated that the transfer and degradation of the investigated pharmaceuticals, these being sulfonamides (sulfanilamide, sulfadiazine, sulfathiazole, sulfameter, sulfadimidine, sulfabenzamide, sulfadimethoxine, and sulfamethoxazole), macrolides (erythromycin, tylosin, roxithromycin), trimethoprim, dicyclanil, penicillin G, carbamazepine, fluvoxamine, and paracetamol, were dependent on the soil characteristics, i.e. on the amount of clay in the soil and its pH. All of the investigated compounds were found to degrade both substantially and rapidly except for roxithromycin and carbamazepine, which were relatively persistent. Kay et al. (2005) investigated the leaching of oxytetracycline, sulfachloropyridazine, and tylosin from clay soils after slurry application. Although the pH was significantly affected by the slurry this had no effect on oxytetracycline leaching. Scheytt et al. (2006) found that diclofenac, ibuprofen, and propyphenazone showed similar mobilities in both saturated and unsaturated column experiments, but carbamazepine showed lower sorption and elimination under unsaturated conditions than under saturated conditions.

### 3.2.2 Pesticides

In contrast to column experiments on pharmaceuticals, most column experiments on pesticide leaching have been carried out under unsaturated conditions in order to reflect the main input path of pesticides into groundwater, which is through agricultural use.

Nkedi-Kizza et al. (1987) carried out leaching experiments on atrazine and diuron using various mixtures of water and methanol. They found a significant reduction in the retardation factor as the volumetric fraction of the organic cosolvent methanol increased. Persson et al. (2008) reported leaching of 30% of the investigated chlorophenols from contaminated soils, of which 1-3% was associated with colloids. Increasing the pore water velocity had no influence on their mobility. Lopez-Blanco et al. (2005) investigated the transport of endosulfan under unsaturated conditions. They found that high soil moisture





favored the transport of this compound by forming and maintaining preferential flow paths in the soil. Rodriguez-Cruz et al. (2007) compared the leaching and retention of linuron, atrazine, and metalaxyl from clayey soils, with and without cationic surfactant treatment. They found that linuron was immobilized in treated soil and that the leaching of atrazine and metalaxyl was reduced. De Wilde et al. (2009) carried out sorption and degradation experiments on pesticides under unsaturated

conditions. The mobility of the investigated pesticides was ranked on the basis of their results as bentazone>metalaxyl>isoproturon>linuron, and their degradability as linuron>metalaxyl>isoproturon>bentazone. Bertelkamp et al. (2014) carried out column experiments on atrazine under saturated conditions and found it to be persistent.

### 3.2.3 Other organic compounds

Persson et al. (2008) investigated the leaching of polychlorinated diphenyl ethers (PCDEs), polychlorinated dibenzofurans

(PCDFs), and polychlorinated dibenzo-p-dioxins (PCDDs) from contaminated soil. Less than 0.2% of PCDEs, PCDFs and PCDDs were mobilized during the experiments and the compounds were found to be preferably associated with the particulate fraction of the leachate. Sinke et al. (1998) investigated the degradation of 4-nitrobenzonate and toluene within a column under a fluctuating water table and reported that both compounds were degraded. They also found that microbial processes induced chemical and physical heterogeneity in the column and that the fluctuating water table introduced additional heterogeneity.

Patterson et al. (2010) conducted column experiments on a number of organic micropollutants under saturated conditions. They found rapid degradation for bisphenol A, 17b-estradiol, 17a-ethynylestradiol, and iodipamide, but only relatively gradual degradation for nitrosodimethylamine, N-nitrosomorpholine and iohexol. Alotaibi et al. (2015) investigated the transport of benzotriazole and 5-methylbenzotriazole under anaerobic saturated conditions and observed biodegradation in both compounds. Liu et al. (2008) simulated the use of bacteria for in situ remediation of biodiesel contamination, using a saturated

column experiment and were able to demonstrate successful degradation of the biodiesel.

### 3.2.4 Non-organic compounds

Although not the primary subject of this review, column experiments are also used to investigate non-organic compounds and selected investigations are presented in this section. Smith et al. (1985) investigated the transport of Escherichia coli through both disturbed and undisturbed soil columns. They found mixed and repacked soils to be much more effective in filtering the

bacteria than undisturbed soils, which allowed up to 96% of the Escherichia coli to pass through the columns. Jin et al. (2000) investigated virus removal and transport in both saturated and unsaturated sand columns and found significantly higher removal under unsaturated flow conditions. Pang et al. (2002) investigated the effect of pore-water velocity on the transport of Cd, Zn, and Pb under non-equilibrium chemical conditions in alluvial gravel columns; they found the proportion of exchange sites available to be independent of the pore-water velocity. Amos et al. (2004) investigated the remediation of acidic mine drainage

using column experiments and found Fe removal during long term operation of column experiments to be a good indicator of the column's ability to remediate acidic mine drainage. Ilg et al. (2007) investigated colloid transport in unsaturated soil columns. They found that colloid transport could be overestimated, depending on the sampling system used.





### 3.3 Comparison of column experiments with other available methods

As described and discussed in Sections 3.1 and 3.2, laboratory column experiments are suitable (and widely used) for investigations into the fate of organic micropollutants. There are, however, alternative methods available that can also be used for this purpose, depending on the objectives and the facilities available:

- Incubation experiments (such as batch experiments, microcosms, etc.) used, for example, to determine sorption coefficients (Scheytt et al., 2005) or to investigate the persistence of pharmaceuticals (Lam et al., 2004), the elimination of pharmaceuticals (Radke and Maier, 2014), the microbial degradation potential of pollutants (Barra Caracciolo et al., 2013), or the biotransformation of micro-contaminants (Nödler et al., 2014).

- Reactive field tracer tests, e.g. on organic micropollutants and pharmaceuticals (Hillebrand et al., 2015; Kunkel and
Radke, 2011; Riml et al., 2013).

By far the largest number of comparisons have been between batch experiments and column experiments on organic micropollutants and we therefore focus on comparisons between these two laboratory methods. However, selected alternative methods are also discussed briefly in this section.

Maeng et al. (2011) investigated the biodegradation of pharmaceuticals in batch and column experiments. More specifically,
the batch experiments were used to investigate removal of the pharmaceuticals and the column experiments were then used to differentiate between biodegradation and sorption of the compounds. Murillo-Torres et al. (2012) used batch and column experiments to investigate the sorption and mobility of organic micropollutants. They obtained contradictory results from the different methods concerning the mobility of di-2-ethyl(hexyl)phthalate and 4-nonylphenol and suggested that the higher mobility of di-2-ethyl(hexyl)phthalate in the column experiments could be due to the formation of complexes within the soil.
Salem Attia et al. (2013) used batch and column experiments to investigate the adsorption of pharmaceuticals to nanoparticles. The batch experiments were used to investigate the influence that the contact time, pH, and concentrations of the compounds had on adsorption and the column experiments were used to investigate the removal efficiency of the nanoparticles. Simon et al. (2000) investigated the influence of redox zonation on the transformation of p-cyanonitrobenzene. They observed a reduction of the compound during column experiments that was an order of magnitude faster than predicted from the results
of previously conducted batch experiments. De Wilde et al. (2009) investigated the sorption and degradation of isoproturon, bentazone, metalaxyl, and linuron. The distribution coefficients that they fitted to the column experiments were much smaller than those obtained from previous batch experiments.

A number of investigations have also combined batch and column experiments in order to optimize their results, for example when investigating the degradation of sulfamethoxazole (Baumgarten et al., 2011), the transport of benzotriazole and its
sorption to zerovalent iron (Jia et al., 2007), or the influence of ozonation on the formation and removal of carbamazepine and its degradation products (Hübner et al., 2013). Ke et al. (2012) used column and batch experiments to investigate the sorption and biotransformation of six endocrine-disrupting compounds (estrone, 17β-estradiol, estriol, 17α-ethynylestradiol, 4-tert-octylphenol, and bisphenol A) and two pharmaceuticals (ibuprofen and naproxen).



Benker et al. (1998) compared the retardation coefficients for trichloroethene estimated from batch and column experiments with field data. They were able to confirm the sorption behavior observed in column experiments from the field data and, provided the sorptive properties of the sediment were correctly determined, batch experiments then allowed the retardation of trichloroethene in the field to be reliably predicted. Scheytt et al. (2007) compared the results obtained from unsaturated column

experiments on clofibric acid, diclofenac, ibuprofen, and propyphenazone to field measurements from former sewage farms. They were thus able to confirm the transport behavior observed in the laboratory experiments through their own field measurements. Bertelkamp et al. (2012) also compared the degradation of organic micropollutants in column experiments with field measurement. Results from laboratory experiments and from investigations at a riverbank filtration site both showed that the charge and Log D of the micropollutants defined their biodegradation potential.

Batch experiments (i.e. incubation experiments in general) can therefore be a useful laboratory method to combine with column experiments in order to characterize the transport behavior of organic micropollutants. Batch experiments can yield reasonable retardation predictions for organic micropollutant compounds. However, sorption coefficients obtained from batch experiments are often not suitable for determining solute transport, either in column experiments or in the field (e.g. De Wilde et al., 2009). This is because in batch experiments sorption is determined under equilibrium conditions, whereas column

experiments determine sorption under non-equilibrium conditions, or at least under dynamic conditions. Sorption also occurs much more rapidly under batch conditions than under flow conditions, which may be due to vibration and a high solution-to-soil ratio (Kookana et al., 1992).

The equilibrium established within a column may differ from that in the inflowing water during the experiment. Batch experiments are often carried out using unrealistic sediment-to-water ratios (e.g. 1:5) that do not reflect realistic aquifer

conditions. While the theoretical maximum sorption capacity (under ideal conditions and equilibrium) can be reasonable well determined from batch experiments, neither advection nor the (dynamic) sorption-desorption behavior (for example) can be determined with this method. Batch experiments are therefore an established method and are suitable for determining equilibrium parameters for interactions between a specific organic and a specific sediment, but they are less able to reproduce the dynamic (i.e. non-equilibrium) groundwater conditions of an aquifer than column experiments. Batch sorption experiments

are therefore suitable for determining the sorption behavior of specific compound-sediment combinations under equilibrium conditions, while column experiments are more suitable for determining the transport behavior of specific compound-fluid-sediment combinations, under either equilibrium or non-equilibrium conditions.

### 3.4 Summary and discussion

The practical objectives of column experiments on organic micropollutants relate to possibilities for their removal, either by

natural processes during passage through soil and groundwater, or by technical processes such as those used in WWTPs. They can also improve our understanding of the relationship between the properties of specific compounds and the properties of fluids and aquifers, with the objective of using organic micropollutants as indicators of aquifer conditions and groundwater history. Different boundary conditions (such as the redox conditions and the degree of saturation) clearly have a strong





influence on the transport and degradation of organic micropollutants, and are therefore critical to defining the transport behavior of a specific organic compound in a way that is applicable to any given hydrogeochemical environment. Unfortunately the results of most column experiments therefore remain restricted to the specific boundary conditions of each column experiment, since variations in just one of these boundary conditions (e.g. the redox conditions) can have a major

impact on the results. For example, the degradation of a compound such as sulfamethoxazole can range from rather low levels (e.g. Suarez et al., 2010) to high levels (e.g. Banzhaf et al., 2012), depending mainly on the redox conditions. The degradation of sulfamethoxazole has been shown to be strongly dependent on the organic carbon content of the sediment (Hebig et al., submitted). The behavior of such compounds can also depend on the level of saturation. Diclofenac, for example, shows high levels of degradation under unsaturated conditions but very low levels of degradation under saturated conditions (Scheytt et

al., 2004; Scheytt et al., 2007).

Another problem faced in designing technical processes to remove organic compounds is that different organic compounds can be sensitive to different hydrochemical conditions, which makes it difficult to find conditions that will allow all such compounds to be removed at the same time. However, column experiments can assist in finding such conditions as it is rather easy with this experimental setup to vary the boundary conditions. Batch sorption experiments are suitable if only the

equilibrium sorption behavior of a specific compound-sediment combination is to be determined. It should be noted that column experiments always remain limited in their transferability to real world conditions because of experimental restrictions (such as those imposed by limitations of scale), which mean that processes that might occur simultaneously in nature cannot be fully reproduced in a laboratory. Column experiments therefore sometimes represent field conditions quite well and sometimes do not. However, the main objective of column experiments should not be specifically to achieve laboratory results

that are transferable to real world conditions but to achieve an improved general understanding of the behavior of organic compounds, i.e. how different boundary conditions affect the behavior of the investigated compounds in natural environments.

**4 Conclusion**

Laboratory column experiments are a valuable and appropriate method for investigating and characterizing the transport behavior of organic micropollutants. They have been widely used in recent decades for numerous investigations into a great

variety of organic compounds. This has led to an enormous increase in our understanding of this ever growing group of compounds. While the experimental method in general can now be considered a standard method, many different setups have been used which is a major issue when it comes to comparing results. A standardized setup for column experiments would yield results that are fully comparable and transferable between different column experiments. It is of course not surprising that such a standardized setup does not exist as the setup used invariably depends very much on the specific research question

being investigated. Steps towards the standardization of column experiments could include following the suggestion by Mackay and Seremet (2008) that model substances be used for investigations into cation exchange, and the use of reference soils to characterize different compounds as suggested by (Bi et al., 2006). These suggestions have to date only been applied





to batch sorption experiments and not to column experiments. It would however be of great benefit if the research community could agree on specific reference compounds and substrates to use in column experiments, in order to help overcome the issue of comparability. This would not only facilitate comparisons between different experiments but also eventually bring us closer to achieving a more universal understanding of the transport and eventual fate of organic micropollutants in groundwater.

Column experiments can provide good estimates of almost all relevant transport parameters for a specific compound in a specific sediment-groundwater setting, such as the retardation (transport velocity compared to groundwater flow velocity), the underlying sorption-desorption processes, and the degradation (as mass loss). However, the results obtained will almost always be limited to the scale of the experiment, which means that they are unlikely to be directly transferable to a field scale as too many parameters will be exclusive to the laboratory column setup. The remaining challenge is therefore to develop

standardized column experiments for organic micropollutants that will be able to overcome this issue.

**Acknowledgements**

The authors thank Martin Gitter for helping to compile the supplementary table. This publication received financial support by the Carl-Fredrik von Horns Fund.

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
