# Peer review of "Table S1: Saturated / flowthrough column experiments"

_Hydrology and Earth System Sciences, 2016_

## Referee Comment (RC1) · Anonymous Referee #1 · 31 May 2016

Overall this paper is fine and I have relatively minor suggestions and corrections below. The paper covers a lot of material outside my specific knowledge area so I cannot assess how truly comprehensive it is, but you will note that one of my comments suggests that the authors are missing some important areas that I am aware of relating to transport of organic compounds that cannot be well described by the equations and models presented here. While I do not expect a comprehensive coverage of this and it would perhaps be too long and deviate from the more focused and specific scope of this work, I do believe that some mention of it in the text, perhaps the conclusions or end of the introduction is warranted as it is important that people understand the limitations not just of column experiments, but of the very models used to interpret the

data.

Comments

Page 4, L 20 – The reactions mentioned there are not what I would say change the rate of transport, but rather change the makeup of the porous medium, which in turn causes transport properties to change.

Figure 1 – I would strongly suggest than in pace of a cartoon , which it appears the authors have created, that the actual curves be plotted as solutions to the advection dispersion reaction equation. The reason I say this is that I am not convinced that the curves are equally comparable – for example it looks like the red one has undergone less dispersion than the blue, but it is further on in the column, which does not make sense unless you explain differences carefully. The curves as drawn qualitatively capture the mechanics, but would better convey them if they were also physically consistent with model predictions.

Page 6, L 15 – only under equilibrium assumption. If the solute is pumped through more quickly than sorption can take place this is not true.

Equation (3) – I don't believe that rho and theta have been defined.

First line Page 8 – degradation can include much more than this – seems way to specific to me.

Page 12 Line 34 – 'may lead to lower flow velocities. . .' I agree that the surface area will be different but lower flow velocities does not make sense. A well designed experiment will try to match dimensionless numbers (Peclet, Damkohler, Reynolds at least). I did not get a strong sense of this from the manuscript and this needs to be much clearer. An experiment that does not at least try to capture and match such quantities will have little relation to a real system, even if flow speed is matched. Dimensionless science is poorly understood and massively underutilized in column experiments and warrants discussion.

This last point is very important relative to one of the discussions the authors have on flow speeds in columns. Yes, it is true that if you have a high flow rate you can conduct a lot of experiments, but the information that you will obtain may be next to useless if the dimensionless numbers do not match, particularly if chemical reactions are involved since systems with high Peclet or Damkohler numbers can behave in fundamentally different ways than those with lower counterparts. Likewise Reynolds numbers can strongly influence the nature and structure of pore scale flows, which can strongly impact larger scale reactive transport.

Page 19, L 9. You mention field experiments, but then it gets no real focus. I would remove this altogether as the focus of this paper is column experiments and ultimately how such information can inform us on field like conditions.

Finally, one thing that is not touched upon at all is that, unlike many inorganic compounds, when discussing organic compounds these can often be made up of a mixture of molecules. The most classical example of this is Natural Organic Matter, which is ubiquitous in natural waters and is by its very definition a mixture of organic compounds of varying molecular size. This is an extreme example, but many other organic compounds have similar mixture properties. The reason I mention this is that when there is interest in understanding the transport of such substances models like the advection dispersion equation are not representative of the transport that actually occurs, even if highly uniform porous materials. The heterogeneity of the chemical properties of the mixture leads to so called anomalous transport behavior, which is due to fractionation, and has been documented in a variety of papers. The three papers that I am most familiar with that relate directly to column experiments are

- Dietrich, Lindsay A. Seders, et al. "Effect of polydispersity on natural organic matter transport." Water research 47.7 (2013): 2231-2240.

- McInnis, D.P et al, 2014. Natural organic matter transport modeling with a continuous time random walk approach. Environmental engineering science, 31(2), pp.98-106.

- McInnis, Daniel P\et al. "Mobility of Dissolved Organic Matter from the Suwannee River (Georgia, USA) in Sand-Packed Columns." Environmental engineering science 32.1 (2015): 4-13.

There are several more, including in the references of the above articles. While the ADE is by far the most used equation to model transport in columns and I strongly believe that it is also one of the most useful, it is important that the community be aware of its limitations. It is not value when considering strongly chemically heterogeneous compounds and given that this can be true for many organic compounds I believe that mention of this is warranted. The ADE still provides useful information, but fails to model certain aspects accurately and such deficiencies can amplify when predictions are made at even larger scales.

---

## Author Comment (AC1) · 12 Jul 2016

Stefan Banzhaf and Klaus H. Hebig

stefan.banzhaf@gu.se

We appreciate the comments provided by Referee #1 on our manuscript and will first provide a reply to the more general comments and then one-by-one replies to the specific comments.

Reply to general comments

The referee claims that we are missing "some important areas [. . .] relating to transport of organic compounds that cannot be well described by the equations and models presented here ". We do, however, think that this does not apply too much to the topic of this manuscript as the literature suggested by the referee focusses on natural

organic matter (NOM) and not so much on organic micropollutants, which are the topic of this review manuscript. The "mixture of molecules" that organic compounds can be made of is mostly not relevant for most laboratory column experiments as they do not investigate complex mixtures but usually single mechanisms. We do agree that complexity is a problem but we do not see that "many other organic compounds have similar mixture properties" as NOM, which is suggested by the referee without providing concrete examples. We therefore do not think that this topic needs to be brought up in this manuscript as NOM is simply out of our review focus, i.e. column experiments on organic micropollutants. Since we do not discuss "strongly chemically heterogeneous compounds" (i.e. NOM) we do not see a need for including the suggested material here.

Reply to specific comments

Comment 1: Page 4, L 20 – The reactions mentioned there are not what I would say change the rate of transport, but rather change the makeup of the porous medium, which in turn causes transport properties to change.

Reply: If the solubility of a compound is changing due to changing parameters, as ORP, T, etc., the compound may sorb or desorb. This does truly change the compound's rate of transport, but not necessarily the aquifer's properties (e.g. the porosity). In most cases the aquifer properties should not significantly be altered due to change of solubility, or redox- or other reactions of organic micropollutants, as they occur typically in the ng to $\mu$g-range.

Comment 2: Figure 1 – I would strongly suggest than in pace of a cartoon, which it appears the authors have created, that the actual curves be plotted as solutions to the advection dispersion reaction equation. The reason I say this is that I am not convinced that the curves are equally comparable – for example it looks like the red one has undergone less dispersion than the blue, but it is further on in the column, which does not make sense unless you explain differences carefully. The curves as

drawn qualitatively capture the mechanics, but would better convey them if they were also physically consistent with model predictions.

Reply: The curves presented in Figure 1 are indeed modelled using the advection dispersion reaction equation, but are plotted at a constant distance (outlet of the column) and shifted for better illustration of the process. However, a strong retardation can deform the curve in a way that dispersion might look bigger than it actually is (i.e. that it looks as if the red curve has undergone less dispersion than the blue one). This is one reason for applying conservative tracers; since they allow for a separation of this effect. To avoid unclear interpretation of the figure we remodeled the figure and plotted the concentrations at a distinct time point (time step 100). The transport parameters used for the forward modelling of the curves are added to the caption: "Figure 1: Schematic representation of solute transport in groundwater, taking into account the main transport processes of advection, hydrodynamic dispersion, retardation, and degradation. BTCs were modelled using the CXTFIT code (Toride et al., 1999). Model was setup as Deterministic equilibrium CDE with flux-averaged concentration and dimensionless parameters. Characteristic length was set as 100. The initial values are: $v = 1$, $D = 0.1*10\text{-}7$, $R = 1$, $\mu = 0$ (grey box, only advective transport); $v = 1$, $D = 15$, $R = 1$, $\mu = 0$ (red curve, advective + dispersive transport); $v = 1$, $D = 15$, $R = 3$, $\mu = 0$ (blue curve, retarded transport); $v = 1$, $D = 15$, $R = 3$, $\mu = 0.01$ (green curve additional degradation). Breakthrough was modelled as multiple pulse input with a concentration of $c = 1$ between time step 10 to 50. The position of the curves within the column are plotted for the time step 100. "

Comment 3: Page 6, L 15 – only under equilibrium assumption. If the solute is pumped through more quickly than sorption can take place this is not true.

Reply: The referee is right about this and we will change the text in this paragraph accordingly: "Under sufficiently low flow rates, equilibrium conditions between solid and fluid phase will establish. Then, a compound can only break through when all sorption places are filled according to the new equilibrium. When the system is flushed

with compound-free water the opposite process takes place, the equilibrium shifts back, and the sorbed compounds are again released into the solution. The result is a delayed breakthrough curve at the observation point (blue curve in Fig. 1). However, under high flow regimes within the column, non-equilibrium conditions might prevail, which can significantly affect the described processes. "

Comment 4: Equation (3) – I don't believe that rho and theta have been defined.

Reply: Rho and theta were defined in the text above the equation as bulk density and porosity. We will put the symbols in brackets for better identification: "If the distribution coefficient between a solid and liquid phase (Kd) of a specific compound is known, together with the bulk density (rho; " symbol will be added in the document as not supported here") and the porosity (Θ) of the substrate, the retardation factor of this compound can be approximated as follows (Stumm and Morgan, 1996): "

Comment 5: First line Page 8 – degradation can include much more than this – seems way to specific to me.

Reply: According to our knowledge the definition of degradation provided here is well describing the processes discussed in this manuscript.

Comment 6: Page 12 Line 34 – 'may lead to lower flow velocities...' I agree that the surface area will be different but lower flow velocities does not make sense. A well designed experiment will try to match dimensionless numbers (Peclet, Damkohler, Reynolds at least). I did not get a strong sense of this from the manuscript and this needs to be much clearer. An experiment that does not at least try to capture and match such quantities will have little relation to a real system, even if flow speed is matched. Dimensionless science is poorly understood and massively underutilized in column experiments and warrants discussion. This last point is very important relative to one of the discussions the authors have on flow speeds in columns. Yes, it is true that if you have a high flow rate you can conduct a lot of experiments, but the information that you will obtain may be next to useless if the dimensionless numbers do not

match, particularly if chemical reactions are involved since systems with high Peclet or Damkohler numbers can behave in fundamentally different ways than those with lower counterparts. Likewise Reynolds numbers can strongly influence the nature and structure of pore scale flows, which can strongly impact larger scale reactive transport.

Reply: We oppose to the first statement as according to Darcy's law high effective porosities do result in lower flow velocities when the same hydraulic gradient is applied. We agree, however, that one could adapt the flow velocity by increasing the hydraulic gradient (i.e. the pumping rate in a column experiment) to achieve a higher flow velocity. As the referee points out, dimensionless science is poorly understood and hence not much is found on this in the literature on column experiments on organic micropollutants (which is what this review is about) and we therefore did not include this in our review. Furthermore, we do not argue that high flow velocities are desirable in column experiments, we state the opposite (p. 13 lines 5-8): "The flow velocity should ideally reproduce natural groundwater flow velocities, which one would normally expect to be between 1 cm d-1 and 1 m d-1. Using higher velocities allows experiments to be completed more quickly and hence many repetitions, but slow velocities are more likely to provide a realistic representation of natural processes, involving equilibration of solute and solid phases. "

Comment 7: Page 19, L 9. You mention field experiments, but then it gets no real focus. I would remove this altogether as the focus of this paper is column experiments and ultimately how such information can inform us on field like conditions.

Reply: We agree that field methods can be excluded here as we do not further discuss them in the manuscript and will therefore remove the respective text passage.
* * *
[Figure]

[Figure]

**Fig. 1.**

---

## Referee Comment (RC2) · Anonymous Referee #2 · 29 Jul 2016

The paper is publishable in its present form with only some slight corrections (here after) The only important recommendation is as follows

I encourage the authors, at the end of the detailed and very analytical description of the observations of column tests on different chemicals to resume, in some way, the main results. For example: which are the main conclusions regarding pharmaceuticals? and regarding pesticides? also if each molecule beahaves in a different way in different settings, is it possible to put in evidence some general rules or behaviours, useful for interested Readers Otherwise the paper seems simply a cold and anlytical list of references and results without any "soul" underlined betwee rows

Here are some specific comments

[Figure]

SPECIFIC COMMENTS

CHAPTER 1 (add question mark to the title) CHAPTER 2:

Add also biodegration, and not only chemical reaction, as a tool of transformation of micropollutants Be careful, because uranine is not a perfectly conservative tracer (sensible to sun light for example and slightly retarded on fine sediments). Only inorganic anions are very near to be conservative. Hydrodynamic dispersion leads not only to a broadening of the breakthrough curve at a particular observation point during flow through a porous medium but also to the dilution of the concentration and to the formation of a tailing due to the pore size distribution effects (correct the red line in Fig.1) Also the curve for retardation must be corrected (lowering of concentration and even more pronounced tailing due to flushing effects and consequent desorption)

Pag.6,r.24: I suggest to use the term of "large specific surface" for organic matter

Pag.6,r.32: what's the meaning of the term "zwitter"? Explain now and not after

Pag.7,r.22: Chemdraw is not in the references

Pag.8,r.11: edit in the proper way the notations of ions with apex in the right position

p.24 r.14: the citation of cambridgesoft.com doesn't seem to be in the main text

p.33 r.25: Schirmer, 2008, is not cited in the main text

---

## Referee Comment (RC3) · Anonymous Referee #3 · 10 Aug 2016

**GENERAL COMMENTS**

The paper addresses a comprehensive review of laboratory column experiments, conducted on different organic compounds and under (very) different experimental setups. Although the technicalities on organic micropollutant transport are outside my field of expertise (hence I cannot tell whether the list cited papers is exhaustive), the paper is easy to follow and attempts a useful synthesis of the main literature results. I particularly appreciate the critical discussion and the conclusions on the validity of column experiments for field applications. The authors often raise the point that standardized column experiments would be beneficial to the scientific community, so I think they could be a bit more explicit about operative steps forward to improve the transferability

of results. In the paper, this is done only briefly in the conclusions, and I am wondering whether it would be worth creating a dedicated section, where proposing more solutions to the community.

MINOR POINTS:

I have the impression the abstract may be rather long. Maybe less general details on column experiments (which are of course well developed in the paper) would make it more efficient as an abstract.

Sec 3.1.3 It may be worth pointing out here that if a column experiment is run under non-stationary conditions (which would be typical in field conditions), then the concentration breakthrough curve does not represent the distribution of transit times of the compound, and the mass breakthrough curve should be used instead.

page 15, line 7: "(which represent the flow velocity of the fluid)", I would reformulate into "(which allow inferring the flow velocity of the fluid)"

TECHNICAL CORRECTIONS:

page 2, line 15: "into a problem contaminant"

---

## Author Comment (AC2) · 22 Aug 2016

Stefan Banzhaf and Klaus H. Hebig

stefan.banzhaf@gu.se

We appreciate the comments provided by Referee #1 on our manuscript and will first provide a reply to the more general comments and then one-by-one replies to the specific comments.

Reply to general comments:

The referee asks for "general rules or behaviors" of pharmaceuticals and pesticides for the "interested readers". We really would like being able to provide these general rules. However, as also stated in the manuscript be cannot provide more than we already did in terms of generalization of column experiments due to very diverse behavior of the

discussed organic compounds. The "classical" parameters like pKa, pH, KOW, and organic content of the aquifer material and ratios of them can be used to estimate the behavior of some organic micropollutants (see also the cited papers by Schaffer & Licha, 2014; 2015).

Reply to specific comments:

Comment 1: CHAPTER 1 (add question mark to the title)

Reply: Changed accordingly

Comment 2: CHAPTER 2: Add also biodegration, and not only chemical reaction, as a tool of transformation of micropollutants Be careful, because uranine is not a perfectly conservative tracer (sensible to sun light for example and slightly retarded on fine sediments). Only inorganic anions are very near to be conservative. Hydrodynamic dispersion leads not only to a broadening of the breakthrough curve at a particular observation point during flow through a porous medium but also to the dilution of the concentration and to the formation of a tailing due to the pore size distribution effects (correct the red line in Fig.1) Also the curve for retardation must be corrected (lowering of concentration and even more pronounced tailing due to flushing effects and consequent desorption)

Reply: Added: "Biodegradation and chemical reactions, resulting in oxidation . . ." The curves presented in Figure 1 are indeed modelled using the advection dispersion reaction equation, but are plotted at a constant distance (outlet of the column) and shifted for better illustration of the process. However, a strong retardation can deform the curve in a way that dispersion might look bigger than it actually is (i.e. that it looks as if the red curve has undergone less dispersion than the blue one). This is one reason for applying conservative tracers; since they allow for a separation of this effect. To avoid unclear interpretation of the figure we remodeled the figure and plotted the concentrations at a distinct time point (time step 100). The transport parameters used for the forward modelling of the curves are added to the caption: "Figure 1: Schematic

representation of solute transport in groundwater, taking into account the main transport processes of advection, hydrodynamic dispersion, retardation, and degradation. BTCs were modelled using the CXTFIT code (Toride et al., 1999). Model was setup as Deterministic equilibrium CDE with flux-averaged concentration and dimensionless parameters. Characteristic length was set as 100. The initial values are: $v = 1$, $D = 0.1*10-7$, $R = 1$, $\mu = 0$ (grey box, only advective transport); $v = 1$, $D = 15$, $R = 1$, $\mu = 0$ (red curve, advective + dispersive transport); $v = 1$, $D = 15$, $R = 3$, $\mu = 0$ (blue curve, retarded transport); $v = 1$, $D = 15$, $R = 3$, $\mu = 0.01$ (green curve additional degradation). Breakthrough was modelled as multiple pulse input with a concentration of $c = 1$ between time step 10 to 50. The position of the curves within the column are plotted for the time step 100. " Figure 1 was recalculated and the respective parameters added to the figure caption.

Comment 3: Pag.6,r.24: I suggest to use the term of "large specific surface" for organic matter

Reply: Changed accordingly

Comment 4: Pag.6,r.32: what's the meaning of the term "zwitter"? Explain now and not after

Reply: This was rectified by moving the respective sentence ("The behavior of organic compounds during transport in groundwater is therefore clearly dependent on their speciation neutral, anionic/acid, cationic/basic, anionic-cationic/zwitter-ionic") to the end of the paragraph.

Comment 5: Pag.7,r.22: Chemdraw is not in the references

Reply: Formatting issue of the reference list: the reference is there but not formatted correctly. Wil be corrected.

Comment 6: Pag.8,r.11: edit in the proper way the notations of ions with apex in the right position

Reply: This was messed up during the last formatting and will be corrected.

Comment 7: p.24 r.14: the citation of cambridgesoft.com doesn't seem to be in the main text

Reply: Formatting issue of the reference list: the reference is there but not formatted correctly. Wil be corrected.

Comment 8: p.33 r.25: Schirmer, 2008, is not cited in the main text

Reply: The reference is cited in the main text: due to the layout of the reference list it might look like that "Schirmer 2008" is a separate reference, which is not the case. The reference is: "Strauch, G., Moder, M.,Wennrich, R., Osenbruck, K., Glaser, H. R., Schladitz, T., Muller, C., Schirmer, K., Reinstorf, F., and Schirmer, M.: Indicators for assessing anthropogenic impact on urban surface and groundwater, J. Soils Sediments, 8, 23 -33, 25, 2008."

---

## Author Comment (AC3) · 22 Aug 2016

Stefan Banzhaf and Klaus H. Hebig

stefan.banzhaf@gu.se

We appreciate the comments provided by Referee #3 on our manuscript and will first provide a reply to the more general comments and then one-by-one replies to the specific comments.

Reply to general comments:

The referee suggests being "a bit more explicit about operative steps forward to improve the transferability of results". We would like being able to provide this explicit information by, e.g. suggesting a standard setup to be used. However, as we also state in the manuscript, we can only suggest setting the boundary conditions of the

column according to literature suggestions on (e.g. the ratio of length to diameter of the column) or based on field analogues (i.e. the groundwater flow velocity). We really do understand that it would be desirable for the scientific community if we could provide more explicit suggestions for the way forward. Unfortunately, we cannot provide more specific suggestions on standardized column experiments.

Reply to specific comments:

Comment 1: I have the impression the abstract may be rather long. Maybe less general details on column experiments (which are of course well developed in the paper) would make it more efficient as an abstract.

Reply: We agree that the abstract is rather long. We would, however, like to keep the abstract in its current form as also the "general details" the referee mentions as being lengthy are an essential part of this manuscript.

Comment 2: Sec 3.1.3 It may be worth pointing out here that if a column experiment is run under non-stationary conditions (which would be typical in field conditions), then the concentration breakthrough curve does not represent the distribution of transit times of the compound, and the mass breakthrough curve should be used instead.

Reply: The referee is right about the issue in case a column experiment is run under non-stationary conditions. However, almost all column experiments are run under stationary conditions in order to control the boundary conditions. We will add a sentence that indicates this potential problem when evaluating the breakthrough of a compound.

Comment 3: page 15, line 7: "(which represent the flow velocity of the fluid)", I would reformulate into "(which allow inferring the flow velocity of the fluid)"

Reply: This will be changed accordingly.

TECHNICAL CORRECTIONS:

Comment 4: page 2, line 15: "into a problem contaminant"

[Figure]

Reply: It is unclear what the referee wants to be corrected here.

---

## Referee Comment (RC4) · Anonymous Referee #4 · 26 Aug 2016

The manuscript presented by Banzhaf and Hebig incorporates an intensive literature research and offers a detailed compilation on laboratory column experiments, including fundamental methods as well as applications/results concerning pharmaceuticals and pesticides. The manuscript is well structured and written in a good style. Hence, it is easy to read and to understand. In order to avoid duplications, I skipped remarks reviewer #1 - #3 already made and, consequently, have to add some minor comments only (see below). In my opinion, the publication of this manuscript will be of large benefit for the community.

Page 8, line 12 ... [...] can therefore induce zones with different oxygen concentrations, zones of nitrate-reduction, iron-reduction, sulfate-reduction ... a suggestion for

reformulation: can therefore induce zones characterized by oxic conditions, nitrate-reduction, [. . .]

Page 8, line 13: suggestion: This transient chemical zoning

Page 8, lines 17-24: The entire section 2.5. seems misplaced . . . One could overcome this issue by renaming this sections (e.g. system specific parameters?) and adequately rewriting it.

Page 18. Line s21-32: as this section (3.2.3 Other organic compounds) seems beyond the scope of the paper (dealing with micropollutants), and the value added to the manuscript is rather low, the authors should consider to remove this section

Page 19, line 11: [. . .] have been between –> have been made between?
* * *

---

## Author Comment (AC4) · 26 Aug 2016

We appreciate the comments provided by Referee #4 on our manuscript and provide a reply to the specific comments below.

Reply to specific comments:

Comment 1: Page 8, line 12 . . . [. . .] can therefore induce zones with different oxygen concentrations, zones of nitrate-reduction, iron-reduction, sulfate-reduction . . . a suggestion for reformulation: can therefore induce zones characterized by oxic conditions, nitrate-reduction, [. . .]

Reply: This was changed accordingly.

[Figure]

Comment 2: Page 8, line 13: suggestion: This transient chemical zoning

Reply: This was changed accordingly.

Comment 3: Page 8, lines 17-24: The entire section 2.5. seems misplaced ... One could overcome this issue by renaming this sections (e.g. system specific parameters?) and adequately rewriting it.

Reply: We agree that the section title is not well chosen and changed it to: "Implications for predicting the transport behavior of organic micropollutants".

Comment 4: Page 18. Line s21-32: as this section (3.2.3 Other organic compounds) seems beyond the scope of the paper (dealing with micropollutants), and the value added to the manuscript is rather low, the authors should consider to remove this section

Reply: We agree that this is a bit out of focus so we removed this section as suggested.

Comment 5: Page 19, line 11: [. . .] have been between –> have been made between?

Reply: This was changed accordingly.